# Multicenter Evaluation of Rapid BACpro^®^ II for the Accurate Identification of Microorganisms Directly from Blood Cultures Using MALDI-TOF MS

**DOI:** 10.3390/diagnostics11122251

**Published:** 2021-12-01

**Authors:** Marina Oviaño, André Ingebretsen, Anne K. Steffensen, Antony Croxatto, Guy Prod’hom, Lidia Quiroga, Germán Bou, Gilbert Greub, David Rodríguez-Temporal, Belén Rodríguez-Sánchez

**Affiliations:** 1Department of Microbiology, Complejo Hospitalario Universitario A Coruña, 15006 A Coruña, Spain; Marina.Oviano.Garcia@sergas.es (M.O.); German.Bou.Arevalo@sergas.es (G.B.); 2Department of Microbiology, Oslo University Hospital, P.O. Box 4950 Nydalen, 0188 Oslo, Norway; aingebre@ous-hf.no (A.I.); a.k.steffensen@medisin.uio.no (A.K.S.); 3Faculty of Medicine, University of Oslo, 0372 Oslo, Norway; 4Institute of Microbiology, University Hospital of Lausanne, CH-1011 Lausanne, Switzerland; Antony.Croxatto@chuv.ch (A.C.); Guy.Prodhom@chuv.ch (G.P.); gilbert.greub@chuv.ch (G.G.); 5Department of Clinical Microbiology and Infectious Diseases, Hospital General Universitario Gregorio Marañón, 28007 Madrid, Spain; lidia.quirogam@gmail.com; 6Instituto de Investigación Sanitaria and Hospital General Universitario Gregorio Marañón (IiSGM), 28007 Madrid, Spain

**Keywords:** rapid BACpro^®^ II kit, MALDI-TOF, mass spectrometry, blood culture, rapid identification

## Abstract

The identification of microorganisms directly from blood cultures using MALDI-TOF MS has been shown to be the most impacting application of this methodology. In this study, a novel commercial method was evaluated in four clinical microbiology laboratories. Positive blood culture samples (*n* = 801) were processed using a rapid BACpro^®^ II kit and then compared with the routine gold standard. A subset of monomicrobial BCs (*n* = 560) were analyzed in parallel with a Sepsityper^®^ Kit (Bruker Daltonics, Bremen, Germany) and compared with the rapid BACpro^®^ II kit. In addition, this kit was also compared with two different in-house methods. Overall, 80.0% of the monomicrobial isolates (609/761; 95% CI 71.5–88.5) were correctly identified by the rapid BACpro^®^ II kit at the species level (92.3% of the Gram negative and 72.4% of the Gram positive bacteria). The comparison with the Sepsityper^®^ Kit showed that the rapid BACpro^®^ II kit generated higher rates of correct species-level identification for all categories (*p* > 0.0001), except for yeasts identified with score values > 1.7. It also proved superior to the ammonium chloride method (*p* > 0.0001), but the differential centrifugation method allowed for higher rates of correct identification for Gram negative bacteria (*p* > 0.1). The percentage of accurate species-level identification of Gram positive bacteria was particularly noteworthy in comparison with other commercial and in-house methods.

## 1. Introduction

The rapid identification of microorganisms causing bloodstream infections (BSI) is one of the most impactful applications of Matrix Assisted Laser Desorption/Ionization-Time of Flight Mass Spectrometry (MALDI-TOF MS) in clinical microbiology [1,2]. This technology has demonstrated the timely and accurate identification of a wide variety of microorganisms directly from positive blood cultures (BCs) and has the added benefit of being inexpensive and efficient [3]. Most microbiology laboratories worldwide have implemented this approach in order for clinicians to promptly initiate optimal antimicrobial treatment [4], which has been shown to correlate with higher rates of positive outcomes [5].

Different pre-processing methods have been developed for the successful identification of microorganisms directly from positive BCs using MALDI-TOF MS. Most of these focus on isolating the microorganisms present in BCs from blood cells and other contaminants by using either a differential centrifugation method [6] or by employing an improved erythrocyte-lysing procedure using ammonium chloride [7], sulfate dodecyl sodium [8] or saponin [9]. Other studies have reported short-incubations of BC broth on agar plates (for 2 to 6 h) prior to MALDI-TOF MS identification from the thin layer of microorganisms grown on the surface of the plate [10]. This proceeding has allowed for the successful identification of bacteria and has even helped overcome some of the limitations of MALDI-TOF MS: the accurate identification of more than one pathogen in polymicrobial infections and reliable identification of Gram positive microorganisms [11].

Commercial kits have been manufactured for the improved lysis of blood cells, and subsequent recovery of the microorganisms present in positive BCs, facilitating the standardization and implementation of this task in the routine of the clinical microbiology laboratory. The Sepsityper^®^ Kit (Bruker Daltonics, Bremen, Germany) is a CE-IVD labeled and FDA approved kit that is widely used, and numerous studies have reported successful direct identification by using this kit [12,13,14,15,16,17]. Obligate and facultative anaerobic species have also been identified using this kit [12,14]. The Vitek^®^ MS Blood Culture Kit (bioMérieux, Lyon, France) is another commercial kit and is currently RUO labeled [18]. High rates of correct species-level identification of different microorganisms using this kit have been reported [19]. However, peer review publications about this kit are still scarce, and further evaluation may be necessary to learn about its performance.

The rapid BACpro^®^ II (Nittobo Medical Co., Tokyo, Japan) is a relatively new CE-IVD labeled kit, which utilizes a polyallylamine-polystyrene copolymer for the efficient isolation of microorganisms [20]. While a few groups have independently reported its clinical performances [21,22,23], further evaluations from multiple centers are needed to judge whether or not to adopt this kit for routine clinical use. In this study, a multicenter evaluation of the rapid BACpro^®^ II (Nittobo Medical Co., Tokyo, Japan) was conducted in four clinical microbiology labs. The performance of this commercial kit was compared with the Sepsityper^®^ Kit and with two in-house methods: differential centrifugation and lysis of blood cells with ammonium chloride (Figure 1).

## 2. Materials and Methods

The evaluation of the rapid BACpro^®^ II kit (Nittobo Ltd., Tokyo, Japan) was carried out in four research centers: the Microbiology Department from Complejo Hospitalario Universitario A Coruña—CHUAC—(A Coruña, Spain), the Department of Microbiology at Oslo University Hospital—OUH—(Oslo, Norway), the Institute of Microbiology from the University Hospital of Lausanne—CHUV—(Lausanne, Switzerland) and the Clinical Microbiology and Infectious Diseases Department from the Gregorio Marañón University Hospital—HGM—(Madrid, Spain). In each laboratory, 200 consecutive positive BCs (from 200 (CHUAC), 200 (OUH), 200 (HGM) and 201 (CHUV) patients) were collected between January and June 2019. Aerobic and anaerobic BC bottles (BD BACTEC Plus Aerobic/F and Plus Anaerobic/F Culture Vials, Becton Dickinson, Franklin Lakes, NJ, USA) were incubated at 35 °C for up to 5 days in the BACTEC™ FX incubation system (Becton Dickinson). When the bottles flagged positive, Gram staining was performed, and a small amount of broth was cultured on suitable agar media and further incubated at 35 °C overnight. Growth on agar media and the subsequent identification by MALDI-TOF MS was considered as a gold standard. In parallel, the evaluation of the rapid BACpro^®^ II kit was carried out.

The performance of the rapid BACpro^®^ II kit (Nittobo Ltd.) was compared with that of the Sepsityper^®^ Kit (Bruker Daltonics, Bremen, Germany) at three centers (CHUAC, HGM and OUH, *n* = 600) At one center (CHUV), the rapid BACpro^®^ II kit was compared with an in-house processing method [7] which employs ammonium chloride (*n* = 201). In addition, the HGM laboratory also compared the rapid BACpro^®^ II kit with their in-house differential centrifugation method (*n* = 200) [2,6] (Figure 1). The identifications obtained by the above-mentioned methods were compared to those attained from colonies grown after culturing the broth from the same BCs on Columbia blood-agar plates (bioMérieux, Lyon, France). The colonies were identified using on-plate protein extraction with 70% formic acid [1].

### 2.1. Sample Preparation Using the Rapid BACpro^®^ II Kit

Bacterial pellets from positive BCs were processed following the manufacturer’s instructions. Briefly, 1 mL of broth was mixed with 500 μL of lysis buffer, and the mixture was centrifuged for 3 min at 2000–5000× *g*. After the supernatant was thoroughly removed, the pellet was resuspended in 800 μL of deionized water and transferred to another tube containing a mixture of 200 μL of cationic polymer and 200 μL of reaction buffer. The mixture was vortexed and centrifuged at 2000–5000× *g* for 1 min. The supernatant was then removed again, and the pellet was resuspended in 800 μL of 70% ethanol. After complete resuspension of the pellet by pipetting it up and down, the sample was centrifuged again in the above-mentioned conditions, and the pellet was submitted to a protein extraction step with 30 μL of formic acid 70% and the same amount of acetonitrile. After another centrifugation step, 1–2 μL of supernatant was deposited on the MALDI target plate, allowed to dry and covered with 1 μL of HCCA matrix, which was prepared according to the manufacturer instructions (Bruker Daltonics, Bremen, Germany).

### 2.2. Sample Preparation Using Sepsityper^®^ Kit

Sample preparation of bacterial pellets was performed according to the instructions of the manufacturer (Bruker Daltonics, Bremen, Germany). The method consisted of mixing 1ml of BC broth with 200 μL of lysis buffer. The mix was centrifuged for 2 min at 13,000 rpm. The pellet was washed with 1ml of washing buffer and centrifuged for 1 min at 13,000 rpm. The supernatant was removed and the pellet submitted to protein extraction with formic acid and acetonitrile. As explained above, 1–2 μL of supernatant was transferred onto the MALDI target plate and covered with 1 μL of HCCA matrix.

### 2.3. Ammonium Chloride Erythrocyte-Lysing Procedure

Ammonium chloride erythrocyte-lysing procedure was applied as previously described [7]. Five ml of positive BC broth was resuspended in 45 mL of deionized water and centrifuged at 1000× *g* for 10 min at room temperature. The supernatant containing lysed blood-cells was discarded, and the pellet was suspended in 1 mL of ammonium chloride (0.15 M NH_4_Cl, 1 mM KHCO_3_, pH 7.31). A second centrifugation step at 140× *g* for 10 min was applied and the supernatant discarded. The final pellet was suspended in 0.2 mL of deionized water and 1–2 μL was transferred to the MALDI target plate and, once dry, covered with formic acid and, subsequently, with HCCA matrix.

### 2.4. Differential Centrifugation Method

From each BC, 8 mL of broth was transferred to a 15 mL tube and centrifuged at low speed (150× *g*) for 10 min. The supernatant was then collected in four 1.5-mL tubes and centrifuged again at maximal speed (13,000 rpm) for 1 min. The pellets from the four tubes were collected in one tube and washed with 1 ml deionized water. After another centrifugation step under the same conditions, the supernatant was discarded and the pellet spotted directly onto the MALDI target plate. The spots were allowed to dry at room temperature and then covered with 1 μL of formic acid. Once dried, HCCA matrix was added [2].

### 2.5. Identification of Microorganisms by MALDI-TOF MS

MALDI-TOF MS analysis was performed on a Bruker Microflex LT mass spectrometer using the MBT Compass Library (#1829023) containing 7331 reference spectra. FlexControl 3.3 and Maldi Biotyper 3.0 software (Bruker Daltonics) were applied to acquire the spectra and for the identification of the isolates with the standard Biotyper module, respectively. The Bacterial Test Standard (Bruker Daltonics) was used for calibration purposes. The identification was performed in duplicates, and the highest score from each pair was recorded.

### 2.6. Interpretation of the Results

In this study, score values ≥2.0 and ≥1.7 were established as the cut-off value for species- and genus-level identification, respectively. Isolates identified with score values below 1.6 were considered only when the first three identifications provided by MALDI-TOF MS were consistent, either at the species- or the genus level. Otherwise, the identification was considered “not reliable”. Agreement at species- and genus level between the methods tested and the gold standard (identification from isolated colonies) was also analyzed.

### 2.7. Statistical Analysis

The sensitivity values for each pre-processing method and 95% confidence intervals were also calculated. They were compared with the sensitivity of the rapid BACpro^®^ II kit using the McNemar test for paired samples with two tails. The validity values were calculated with a 95% confidence interval (CI) following an exact binomial distribution.

## 3. Results

### 3.1. Performance of the Rapid BACpro^®^ II Kit

During the study period, 801 consecutive, positive BCs were collected in the four participating laboratories, and microorganisms present in these cultures were identified by MALDI-TOF MS after sample pre-processing using the rapid BACpro^®^ II kit. In 761 BCs (193 from CHUAC, 190 from OUH, 177 from HGM and 201 from CHUV), only one species was present (Table 1 and Appendix A), while in 27 BCs, more than one microorganism was detected (Appendix A). In the remaining BCs (*n* = 13), no growth was recorded after 24 h incubation on agar plates, despite flagging positive in their BACTEC systems. The Gram staining of these BCs were reviewed and considered as inconclusive in 13/13 cases. Therefore, these BCs were considered as false positives and removed from the study.

The implementation of the rapid BACpro^®^ II kit in the four participating laboratories allowed for an overall 80.0% (609/761; 95% CI 71.5–88.5) correct identification rate at the species level (score values ≥ 2.0) of the pathogens present in the monomicrobial BCs (Table 1). Among the different bacterial groups, this rate ranged between 66.7% and 89.0% (Appendix A). The identification at the genus level (scores ranging between 1.7 and 2.0) was accurate in another 110/761 isolates (14.4%, 95% CI 8.63–20.2). The identification score values were below 1.7 in only 42 cases (5.6–95% CI 2.56–8.64). This group of microorganisms was mainly composed of Gram positives (29/42) and yeasts (8/42), confirming the limitations of MALDI-TOF MS identification when applied directly to BC broth. Despite the low-confidence scores, 100% of these identifications agreed with the gold standard method. Of the Gram negatives, 92.3% were identified with score values ≥ 2.0 (298/323; 95% CI 91.6–93.0). The number of accurate identifications at the species level was above 80.0% for all main groups of bacteria within this category.

### 3.2. Comparison between the Rapid BACpro^®^ II Kit and Sepsityper^®^ Kit

A subset of 560 monomicrobial BCs were analyzed at CHUAC, HGM and OUH (Figure 1) with both the rapid BACpro^®^ II kit and Sepsityper^®^ Kit in a head-to-head evaluation (Table 2). Overall, 76.8% (430/560; 95% CI 67.6–85.4) of the BCs were identified at the species-level with the rapid BACpro^®^ II kit, compared to 69.3% (388/560; 95% CI 62.3–76.3) with the Sepsityper^®^ Kit. A similar number of BCs was identified at the genus level using both kits (16.4% for the rapid BACpro^®^ II kit (92/560; 95% CI 16.3–16.5) compared to 15.2% for the Sepsityper^®^ Kit –85/560; 95% CI 15.1 to 15.3). The rate of unreliable identifications was higher with the Sepsityper^®^ Kit (6.8% (38/560; 95% CI 6.8–6.8) vs. 15.5%−87/560; 95% CI 15.5 to 15.5) (Figure 2). Gram negatives were successfully identified at the species level with both kits (90.5% (210/232; 95% CI 90.5–90.7) for the rapid BACpro^®^ II kit, compared to 85.8% (199/232; 95% CI 85.7–85.9) for the Sepsityper^®^ Kit). Interestingly, differences were observed in the number of unreliable identifications (1.7% (4/232; 95% CI 1.69–1.71) for the rapid BACpro^®^ II kit compared to 10.8% (25/232; 95% CI 10.7–10.9) for the Sepsityper^®^ Kit). Regarding Gram positives, the rate of species-level identifications yielded by the rapid BACpro^®^ II kit was higher (68.7% −215/313; 95% CI 68.6–68.8) than the Sepsityper^®^ Kit (59.7%, −187/313; 95% CI 59.6–59.8), and the number of BCs with score values below 1.7 was reduced (8.3% −26/313; 95% CI 8.26–8.34 vs. 17.9% −56/313; 95% CI 17.9–17.9). The proportion of yeasts identified at the genus level, however, was higher using the Sepsityper^®^ Kit (46.7% (7/15; 95% CI 7.2–86.2)) than the rapid BACpro^®^ II (13.3% (2/15; 95% CI 0–26.6)) and the number of unreliable isolates was lower (40.0% (6/15; 95% CI 0–80) vs. 53.4% (8/15; 95% CI 22.8–84)), although species-level identification was achieved more often with the rapid BACpro^®^ II kit (33.3% (5/15; 95% CI −7.3–73.9)) than the Sepsityper^®^ Kit (13.3% (2/15; 95% CI −36.2–62.8)) (Table 2). Overall, the rapid BACpro^®^ II kit showed a higher sensitivity both at the species- (*p* < 0.0001, OR = 0.429 (95% CI 0.287–0.630) and the genus-level (*p* < 0.0001, OR = 0.227 (95% CI 0.125–0.388).

In this study, we also included 27 polymicrobial BCs (Appendix A). Both commercial kits allowed for the correct species-level assignment of one microorganism in 23/27 BCs with score values ranging between 1.67 and 2.55 for the rapid BACpro^®^ II kit and in 22/27 cases with the Sepsityper^®^ Kit, with scores between 1.75 and 2.45. In two cases, both pathogens present in the same BC could be identified using the rapid BACpro^®^ II kit (*Staphylococcus epidermidis* plus *Staphylococcus capitis* and *Klebsiella pneumoniae* plus *Staphylococcus haemolyticus*) and in a different case with the Sepsityper^®^ Kit (*Staphylococcus aureus* plus *Enterococcus faecium*). Finally, both kits failed to identify all microorganisms from the same BC in two cases (*Capnocytophaga sputigena* plus *Fusobacterium nucleatum* and *S. aureus* plus *Streptococcus mitis*), and the Sepsityper^®^ Kit also failed in two more cases where two microorganisms were present (Appendix A).

### 3.3. Comparison between the Rapid BACpro^®^ II Kit and in-House Methods

#### 3.3.1. The Ammonium-Chloride Method

The performance of the rapid BACpro^®^ II kit was also compared in the CHUV laboratory with their in-house method that employs ammonium chloride for cell lysis [7] in a head-to-head assay (Table 3). This analysis showed that the rapid BACpro^®^ II allowed for the correct identifications at the species level with higher accuracy than the in-house method (89.0% (179/201) vs. 39.8% (80/201); *p* < 0.01) (Figure 3). The rate of the correct species-level assignment of the rapid BACpro^®^ II kit was higher for both the Gram negatives (96.7% (88/91) vs. 53.8% (49/91)) and Gram positives (83.6% (87/104) vs. 26.9% (28/104)). In the first case, the application of the commercial kit allowed for the identification of 97.4% (75/77) of the microorganisms from the Enterobacteriaceae family and 100% (5/5) of the *Pseudomonas*, compared to 61.0% (47/77) and 0% (0/5) when the in-house method was applied. In the case of the Gram positives, the ammonium chloride method allowed for 40.0% (8/20) of the species-level identification of *S. aureus* compared to 100% (20/20) using the rapid BACpro^®^ II kit (Table 3). The same is true for other groups of Gram positive microorganisms (Coagulase negative (CoN)-Staphylococci, Staphylococci and Enterococci) for which the rapid BACpro^®^ II kit yielded a 71.4–100% accurate species-level identification, whilst the ammonium chloride method ranged between 26.4 and 28.6% for correct species-level identification. Similarly, the ammonium chloride method yielded high rates of unreliable identifications for both groups of microorganisms: 19.8% (18/91) of the Gram negatives and 38.5% (40/104) of the Gram positives were identified by MALDI-TOF MS with score values ≤ 1.7, whilst the use of the rapid BACpro^®^ II kit reduced the number of non-reliable identifications to 1.1% (1/91) of the Gram negatives and 2.9% (3/104) of the Gram positives (Table 3). Likewise, for the yeasts, a higher number of isolates were identified at species levels when the commercial kit was applied (66.7% (4/6)) compared to 50.0% (3/6) with the in-house method.

#### 3.3.2. Differential Centrifugation

The performance of the rapid BACpro^®^ II kit was also compared with the differential centrifugation method in the HGM laboratory (Table 4). Overall, the rapid BACpro^®^ II allowed for a 66.7% (118/177) correct species-level assignment compared to the 61.6% (109/177) obtained by using the in-house kit (*p* = 0.14; OR = 1.54 −0.88–2.77 CI 95%) (Figure 4). Differences in the identification of Gram negatives and Gram positives were observed. The rate of correct identifications at the species level for Gram negatives was higher for the differential centrifugation method due to the increased number of isolates from the Enterobacteriaceae (87.0% (60/69)) correctly identified by this method, compared to 84.1% (58/69) with the rapid BACpro^®^ II kit. However, the number of *Pseudomonas aeruginosa* correctly identified at the species level was higher with the commercial kit (100% (4/4) vs. 50.0% (2/4)). Conversely, successful species-level identification was obtained for several Gram positives using the rapid BACpro^®^ II kit, especially for *S. aureus* (91.7% (11/12) vs. 25.0% (3/12)) and Enterococci (91.7% (11/12) vs. 75.0% (9/12)), which added to the global superiority of the commercial kit for the identification of Gram positives at the species level (58.0% (51/88) vs. 44.3% (39/88)), despite its lower rates of correct identifications at this level for Streptococci (36.7% (11/30) vs. 46.7% (14/30)). Finally, the percentage of yeast correctly identified by the rapid BACpro^®^ II kit was also lower (12.5% (1/8) vs. 25.0% (2/8)), but the differential centrifugation method yielded a higher rate of unreliable identified isolates (75.0% −6/8 vs. 62.5% −5/8).

Of the polymicrobial BCs, 23 cases were included in this head-to-head comparison. The rapid BACpro^®^ II kit allowed for the identification of at least one microorganism in 22/23 cases with score values between 1.75 and 2.55, while the differential centrifugation method provided the identification of one microorganism in 20/23 cases, with score values ranging between 1.45 and 2.45. Moreover, the commercial kit allowed for the identification of both pathogens in two cases (*S. epidermidis* plus *S. capitis* and *E. faecalis* plus *S. epidermidis*) and the in-house method in one case (*Enterococcus faecium* and *Aeromonas hydrophila* in a BC where *K. pneumoniae* was also present).

## 4. Discussion

With the advent of MALDI-TOF MS for the rapid identification of microorganisms, several pre-processing methods have been developed for the direct identification of pathogens present in BCs which encompass different in-house methods [2,6,7,11] and three commercial kits: (i) the Sepsityper^®^ Kit [12,15,16] and (ii) the Vitek^®^ MS Blood Culture Kit [19], which are based on the use of a lysis buffer and improved cleansing of the bacteria by centrifugation or filtration, and (iii) the rapid BACpro^®^ II kit [20,23]. The rapid BACpro^®^ II contains a polyallylamine-polystyrene copolymer for the recovery of microorganisms present in BCs. Bacterial isolates have been shown to adhere to positively charged polymers, forming macroscopic aggregates that can be easily recovered by centrifugation [20].

The Sepsityper^®^ Kit has been extensively evaluated and successful, accurate identifications at the species level of over 80.0% have been reported [8,12,13,15,16,17,21]. On the other hand, the performance of the rapid BACpro^®^ II has been reported by only a few groups as single center studies [21,22,23]. Tsuchida et al. [22] analyzed a collection of 193 monomicrobial BCs and demonstrated an overall correct microbial identification of 80.8% and 99.5%, with score values cut-offs at ≥2.0 and ≥1.7, respectively. The same study observed that all Gram positives were correctly identified at the species level with score values ≥1.7 and in 69.4% of the cases with score values ≥2.0. In a more recent study including 199 BCs, Kayin et al. [21] showed that the rapid BACpro^®^ II kit yielded an overall 87.4% correct species-level identification (score value ≥ 1.7) using the standard module of the Biotyper system (Bruker Daltonics). This rate increased to 94.4% when the specific Sepsityper module for Sepsityper Kit (score value ≥ 1.6) was applied. Gram positives were accurately identified at the species level in 91.5% of the cases using the Sepsityper module [21].

In our study, 761 monomicrobial BCs were analyzed in four laboratories using the rapid BACpro^®^ II kit. We identified 80.0% of the isolates correctly at the species level with score values ≥2.0, and 94.4% correctly at the genus level with score values ≥1.7. The remaining isolates (*n* = 42) could be reliably identified despite the lower scores. These results agree with those obtained by Tsuchida et al. (2018) and Kayin et al. (2019). The Enterococci (97.3%) and *S. aureus* (94.7%) were correctly identified with scores ≥2.0 more often than other Gram positives such as Streptococci (66.3%) and coagulase negative Staphylococci (59.5%). Regarding the Gram negatives, 92.3% of the isolates were correctly identified at species level with score values ≥2.0, and 38.1% of the yeasts were unreliably identified even when applying score values <1.7 as the cut-off. Nevertheless, only 42.9% of the 21 yeast samples were accurately identified with score values ≥2.0, and 38.1% of them were unreliably identified even when applying score values <1.7 as the cut-off. While the study by Yonezawa et al. [23] showed high rates of the correct identification of *C. albicans* and *C. parasilosis*, it has been difficult to judge the rapid BACpro^®^ II’s performance on yeast due to the low abundance of clinical samples [21,22,23]. The low capacity of MALDI-TOF MS to identify yeasts from BCs at species level was already reported [13], likely due to the presence of a specific cell wall and the interference of blood components in the protein spectra [24]. With a relatively larger sample size, our study revealed one of the limitations of the rapid BACpro^®^ II kit, which was also reported as a challenge for the Sepsityper Kit [15].

A subset of 560 BCs was analyzed with the rapid BACpro^®^ II and Sepsityper^®^ Kit in a head-to-head comparison. High rates of Gram negatives were correctly identified by both kits: 90.5% with the rapid BACpro^®^ II kit and 85.8% with Sepsityper^®^ Kit (score values ≥ 2.0). The rate of correct species assigned was higher than 90% for *E. coli*, *K. pneumoniae* and other Enterobacteriaceae using both kits, as previously observed [12]. However, all *P. aeruginosa* isolates were identified with score values ≥2.0 using the rapid BACpro^®^ II kit compared to 70.0% with the Sepsityper^®^ Kit. Higher differences were observed for Gram positive bacteria: 68.7% with the rapid BACpro^®^ II kit (score values ≥ 2.0) and 59.7% with the Sepsityper^®^ Kit. Within this group, major differences were observed in the identification of coagulase-negative Staphylococci (55.9% with the rapid BACpro^®^ II kit and 43.2% with the Sepsityper^®^ Kit (score values ≥ 2.0)) and Enterococci (96.7% with the rapid BACpro^®^ II kit and 76.7% with the Sepsityper^®^ Kit (score values ≥ 2.0)). Errors in the identification of coagulase negative bacteria present in BC pellets have been reported [7]. However, no misidentifications with any of the applied methods have been detected in this study, showing that the methodology applied is highly specific.

The performance of two in-house pre-processing methods were also compared with the rapid BACpro^®^ II kit. The method containing a lysis step with ammonium chloride showed lower accuracy than the rapid BACpro^®^ II kit when applied to the different microbial pellets. Species-level identifications for the rapid BACpro^®^ II and the ammonium chloride method were, respectively, 89.0% and 39.8% for the overall, 96.7% and 53.8% for Gram negatives, 83.6% and 26.9% for Gram positives and 66.7% and 50.0% for the yeast. The rates of identification by the ammonium chloride method in this study were lower than those previously obtained by the same research group (83.0% and 42.0% of correct species level identifications for Gram negative and Gram positive bacteria, respectively) [7]. This differences may be due to the diversity of bacterial species included in this study. Regarding the differential centrifugation method, the overall rate and the amount of correct identifications of Gram positives was higher for the rapid BACpro^®^ II kit (58.0%) than the in-house method (44.3%), but the in-house method gave more correct identification of Gram negatives (83.9%) than the rapid BACpro^®^ II (81.5%).

Among the 27 polymicrobial BCs, we could identify both pathogens twice (7.4%) with the rapid BACpro^®^ II kit and once (3.7%) with the Sepsityper^®^ Kit. These rates are lower than those reported by Scohy et al. (2018), who identified two pathogens in 36.8% of the cases by applying the specific Sepsityper module from the Compass IVD software (Bruker Daltonics) [17]. The combination of the improved sample preparation method and the ad-hoc method for spectra acquirement may increase the identification of polymicrobial BCs. In order to elucidate this, further evaluation of the rapid BACpro^®^ II kit performance coupled to the Sepsityper module is desired.

## 5. Conclusions

In summary, although the evaluated rapid BACpro^®^ II kit proved less efficient for the identification of yeasts, it allowed for a high rate of correct identifications of Gram negative and Gram positive bacteria. The results obtained were consistent in the four laboratories where the kit was tested, showing a reduced number of unreliable identifications and a high rate of Gram positive bacteria reliably identified at the species level. Hands-on time was calculated as 30 to 40 min for 1 to 10 BCs, which enables the implementation of this methodology in the laboratory routine in a standardized way. Moreover, the kit is manufactured independently from the mass spectrometry systems, which provides flexibility to the MALDI-TOF MS user regarding the optimized use of available reagents and represents a good alternative to the limited number of available commercial kits. In addition, we confirmed the robustness of the MALDI-TOF MS approach, since we observed no discordant results, even when applying different pellet preparation methods.

## Figures and Tables

**Figure 1 diagnostics-11-02251-f001:**
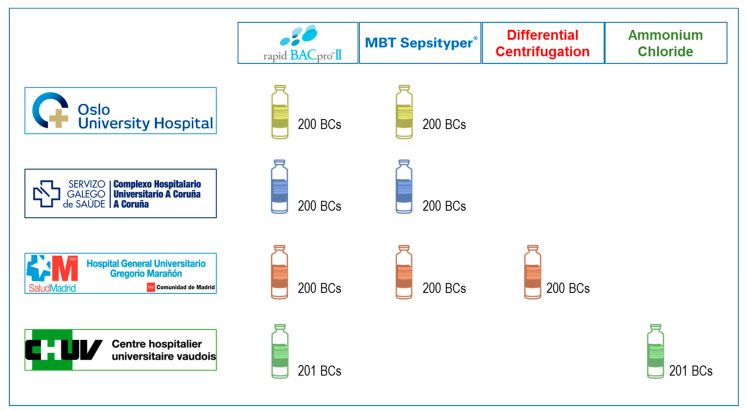
Study design. 200 BCs were collected in each laboratory—201 in the CHUV—and analyzed using the rapid BACpro^®^ II kit. The results were compared with Sepsityper^®^ Kit (CHUAC, HGM and OUH), with the ammonium-chloride lysis system (CHUV) and with the differential centrifugation method (HGM).

**Figure 2 diagnostics-11-02251-f002:**
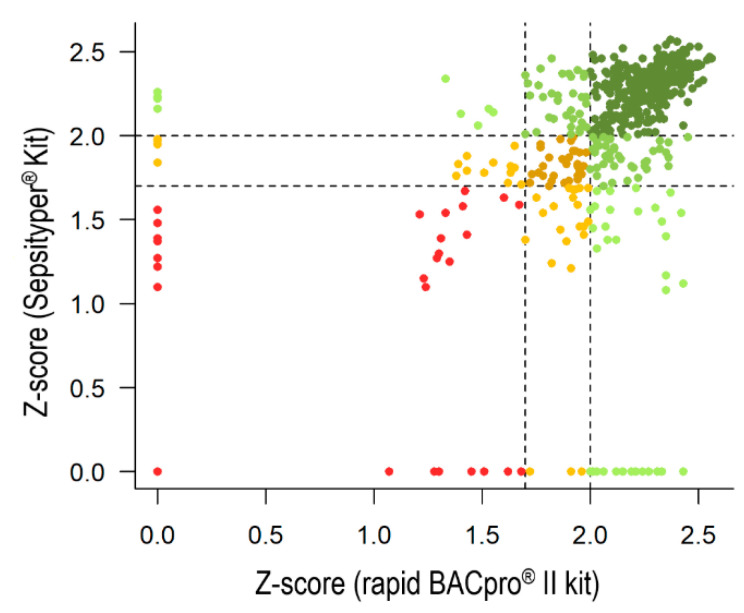
Scattering plot representing the distribution of the scores obtained by a group of 560 monomicrobial BCs analyzed using the rapid BACpro^®^ II (in the x-axis) and the Sepsityper^®^ Kit (in the y-axis). Each spot represents an isolate identified at the species level by both methods (dark green); at species level by one method and at the genus level by the other (green); at the species level by one method and unreliably by the second method (light green); at the genus level by both methods (brown); and at the genus level by one method and unreliably by the second method (yellow). Finally, isolates unreliably identified by both methods are represented in red.

**Figure 3 diagnostics-11-02251-f003:**
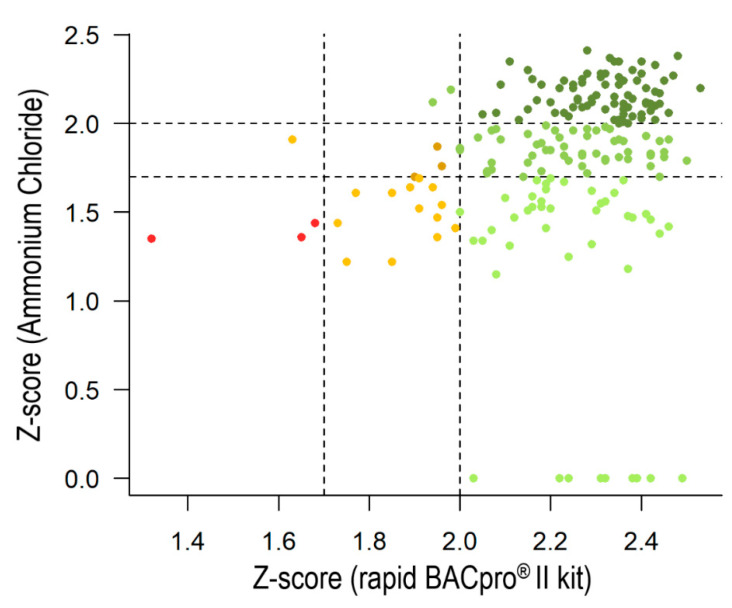
Representation of the scores from 201 BCs analyzed by applying the rapid BACpro^®^ II (in the x-axis) and the ammonium chloride method (in the y-axis). The same color code as in Figure 2 is applied.

**Figure 4 diagnostics-11-02251-f004:**
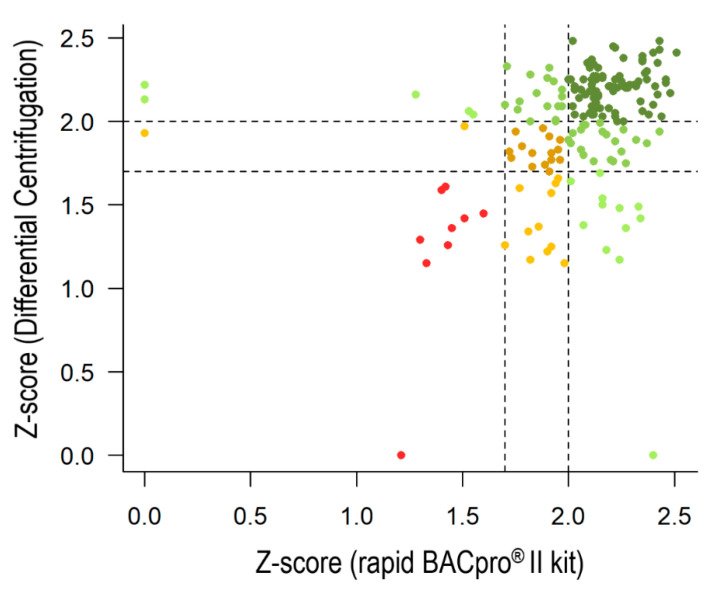
Representation of the scores from 177 monomicrobial BCs analyzed by applying the rapid BACpro^®^ II (in the x-axis) and the differential centrifugation method (in the y-axis). The same color code as in Figure 2 is applied.

**Table 1 diagnostics-11-02251-t001:** Identification by MALDI-TOF MS using the rapid BACpro^®^ II kit of all monomicrobial BCs analyzed in the four participating laboratories.

	*n*	Rapid BACpro II					
		S >2.0	(%)	1.7 < S < 1.99	(%)	S < 1.7	(%)
(A)							
Overall	761	609	80.0	110	14.4	42	5.6
Gram-Negative (GN)	323	298	92.3	20	6.2	5	1.5
Gram-Positive (GP)	417	302	72.4	86	20.6	29	7.0
(B)							
Enterobacterales	271	254	93.7	15	5.5	2	0.8
* Escherichia coli*	149	139	93.3	9	6.0	1	0.7
*Klebsiella* spp.	70	68	97.1	2	2.9	0	0
*Pseudomonas*	15	15	100	0	0	0	0
Other GN	37	29	78.4	5	13.5	3	8.1
*S. aureus*	95	90	94.7	2	2.1	3	3.2
CoN staphylococci	153	91	59.5	53	34.6	9	5.9
Streptococci	86	57	66.3	16	18.6	13	15.1
Enterococci	37	36	97.3	1	2.7	0	0
Other GP	46	28	60.9	14	30.4	4	8.7
Yeast	21	9	42.9	4	19.0	8	38.1

**Table 2 diagnostics-11-02251-t002:** Head-to-head comparison of the performance of both commercial kits on a set of monomicrobial BCs (*n* = 560) analyzed in three laboratories (CHUAC, HGM and OUH).

	*n*	rapid BACpro II						Sepsityper Kit					
		S >2.0	(%)	1.7 < S < 1.99	(%)	S < 1.7	(%)	S >2.0	(%)	1.7 < S < 1.99	(%)	S < 1.7	(%)
(A)													
Overall	560	430	76.8	92	16.4	38	6.8	388	69.3	85	15.2	87	15.5
Gram-Negative (GN)	232	210	90.5	18	7.8	4	1.7	199	85.8	8	3.4	25	10.8
Gram-Positive (GP)	313	215	68.7	72	23.0	26	8.3	187	59.7	70	22.4	56	17.9
(B)													
Enterobacterales	194	179	92.3	14	7.2	1	0.5	179	92.3	5	2.6	10	5.1
* Escherichia coli*	112	103	92.0	8	7.1	1	0.9	105	93.7	2	1.8	5	4.5
*Klebsiella* spp.	43	41	95.3	2	4.7	0	0	41	95.4	1	2.3	1	2.3
*Pseudomonas*	10	10	100	0	0	0	0	7	70.0	0	0	3	30.0
Other GN	28	21	75.0	4	14.3	3	10.7	13	46.4	3	10.7	12	42.9
*S. aureus*	75	70	93.3	2	2.7	3	4.0	68	90.7	2	2.7	5	6.6
CoN staphylococci	118	66	55.9	44	37.3	8	6.8	51	43.2	36	30.5	31	26.3
Streptococci	67	39	58.2	15	22.4	13	19.4	34	50.7	21	31.3	12	18.0
Enterococci	30	29	96.7	1	3.3	0	0	23	76.7	4	13.3	3	10.0
Other GP	23	11	47.8	10	43.5	2	8.7	11	47.8	7	30.4	5	21.8
Yeast	15	5	33.3	2	13.3	8	53.4	2	13.3	7	46.7	6	40.0

**Table 3 diagnostics-11-02251-t003:** Head-to-head comparison between the performances of the rapid BACpro^®^ kit and the in-house method described by Prod’hom et al. (2010) which uses ammonium chloride.

	*n*	Rapid BACpro II						Ammonium Chloride Method					
		S >2.0	(%)	1.7 < S < 1.99	(%)	S < 1.7	(%)	S >2.0	(%)	1.7 < S < 1.99	(%)	S < 1.7	(%)
(A)													
Overall	201	179	89.0	18	9.0	4	2.0	80	39.8	60	29.9	61	30.3
Gram-Negative (GN)	91	88	96.7	2	2.2	1	1.1	49	53.8	24	26.4	18	19.8
Gram-Positive (GP)	104	87	83.6	14	13.5	3	2.9	28	26.9	36	34.6	40	38.5
(B)													
Enterobacterales	77	75	97.4	1	1.3	1	1.3	47	61.0	19	24.7	11	14.3
*Escherichia coli*	37	36	97.3	1	2.7	0	0	26	70.3	5	13.5	6	16.2
*Klebsiella* spp.	27	27	100	0	0	0	0	18	66.7	5	18.5	4	14.8
*Pseudomonas*	5	5	100	0	0	0	0	0	0	2	40.0	3	60.0
Other GN	9	8	88.9	1	11.1	0	0	2	22.2	3	33.3	4	44.5
*S. aureus*	20	20	100	0	0	0	0	8	40.0	9	45.0	3	15.0
CoN staphylococci	35	25	71.4	9	25.7	1	2.9	10	28.6	12	34.3	13	37.1
*Streptococci*	19	18	94.7	1	5.3	0	0	5	26.4	7	36.8	7	36.8
*Enterococci*	7	7	100	0	0	0	0	2	28.6	1	14.3	4	57.1
Other GP	23	17	73.9	4	17.4	2	8.7	3	13.1	7	30.4	13	56.5
Yeast	6	4	66.7	2	33.3	0	0	3	50.0	0	0	3	50.0

**Table 4 diagnostics-11-02251-t004:** Comparison between the performances of the rapid BACpro^®^ test and the in-house Differential Centrifugation on a group of monomicrobial BCs (*n*=177).

	*n*	rapid BACpro II						Differential Centrifugation Method					
		S >2.0	(%)	1.7 < S < 1.99	(%)	S < 1.7	(%)	S >2.0	(%)	1.7 < S < 1.99	(%)	S < 1.7	(%)
(A)													
Overall	177	118	66.7	43	24.3	16	9.0	109	61.6	36	20.3	32	18.1
Gram-Negative (GN)	81	66	81.5	13	16.0	2	2.5	68	83.9	11	13.6	2	2.5
Gram-Positive (GP)	88	51	58.0	28	31.8	9	10.2	39	44.3	25	28.4	24	27.3
(B)													
Enterobacterales	69	58	84.1	10	14.5	1	1.4	60	87.0	8	11.6	1	1.4
*Escherichia coli*	35	28	80.0	6	17.1	1	2.9	31	88.6	4	11.4	0	0
*Klebsiella* spp.	15	14	93.3	1	6.7	0	0	14	93.3	1	6.7	0	0
*Pseudomonas*	4	4	100	0	0	0	0	2	50.0	2	50.0	0	0
Other GN	8	4	50.0	3	37.5	1	12.5	6	75.0	1	12.5	1	12.5
*S. aureus*	12	11	91.7	1	8.3	0	0	3	25.0	2	16.7	7	58.3
CoN staphylococci	27	16	59.3	11	40.7	0	0	12	44.5	9	33.3	6	22.2
Streptococci	30	11	36.7	11	36.7	8	26.6	14	46.7	9	30.0	7	23.3
Enterococci	12	11	91.7	1	8.3	0	0	9	75.0	2	16.7	1	8.3
Other GP	7	2	28.6	4	57.1	1	14.3	1	14.3	3	42.8	3	42.8
Yeast	8	1	12.5	2	25.0	5	62.5	2	25.0	0	0	6	75.0

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
