# Peer review of "Multicenter Evaluation of Rapid BACpro® II for the Accurate Identification of Microorganisms Directly from Blood Cultures Using MALDI-TOF MS"

_diagnostics, 2021, doi:10.3390/diagnostics11122251_

Round 1

Reviewer 1 Report

The authors present a multi-centre head-to-head real life comparison of several methods of fast species and genus identification from blood culture using MALDI TOF. The rapidBAC pro II system was compared against the Sepsityper kit, as well as two in-house methods with identification from colonies on agar plates used as control. All in all, the study is a well-done and useful comparison of several ways of obtaining quick identification from blood cultures. I done have some minor comments that I believe would improve tha mauscript.

  1. Explicitly listing the species/genus of the cultures that none of the systems were able to correctly identify would help showcase the strengths and weaknesses of the rapid systems generally.
  2. Overall sensitivity of the rapidBAC II test (as in Table 1) could be shown with confidence intervals.
  3. It is not apparent that the listing "Enterobacteriales" (should probably be "Enterobacterales") includes E. coli and Klebsiella or if it is only other species in the group. Pseudomonas also seems to be included in Enterobacterales, which it is not a member of. Please consider redesigning the tables to make this clearer.
  4. It is not clear what extraction method, if any, was used for the "gold standard" MALDI TOF from colonies on agar.
  5. How were cases with scores below 1.7 with all the top hits being the same handled? Reported to the genus level?

Author Response

First of all, all authors want to thank the reviewers for their thorough revision of our manuscript.

The text has been revised by a native speaker to improve its clarity.

Please find our answers below.

Reviewer 1

The authors present a multi-centre head-to-head real life comparison of several methods of fast species and genus identification from blood culture using MALDI TOF. The rapidBAC pro II system was compared against the Sepsityper kit, as well as two in-house methods with identification from colonies on agar plates used as control. All in all, the study is a well-done and useful comparison of several ways of obtaining quick identification from blood cultures. I done have some minor comments that I believe would improve tha manuscript.

  1. Explicitly listing the species/genus of the cultures that none of the systems were able to correctly identify would help showcase the strengths and weaknesses of the rapid systems generally.

In the case of the 761 blood cultures where only one microorganism was detected (Table 1), all isolates analyzed were identified with higher or lower score. In this case, the challenge is to know if the identifications obtained with low scores are reliable or not. This fact has been stated in Lines 334-335: “The remaining isolates (n=42) could be reliably identified despite the lower scores. “

Only in BCs where more than one pathogen was present usually one of them was missed by MALDI-TOF (Lines 223-233)

  1. Overall sensitivity of the rapidBAC II test (as in Table 1) could be shown with confidence intervals.

95% Confidence intervals for the data sourcing from more than one laboratory (Tables 1 and 2) have been stated. The expert statisticians from our department do not recommend the calculation of 95% CI for the results of the in-house methods.

  1. It is not apparent that the listing "Enterobacteriales" (should probably be "Enterobacterales") includes E. coli and Klebsiella or if it is only other species in the group. Pseudomonas also seems to be included in Enterobacterales, which it is not a member of. Please consider redesigning the tables to make this clearer.

We agree with the reviewer: the name of the bacterial order was misspelt. It has been corrected in the reviewed version of the manuscript.

Within the Enterobacterales order, we meant to show the results for E. coli and K. pneumoniae since these species are frequently found in BCs. Different alignment for these two categories has been used in Tables 1-4. We never meant to include P. aeruginosa as part of the Enterobacterales order. We do hope this is clear now with the new display used for the tables.

  1. It is not clear what extraction method, if any, was used for the "gold standard" MALDI TOF from colonies on agar.

We missed the explanation about how the reference method (“gold standard”) for MALDI-TOF identification was performed. The details about it are stated in lines 102-105: “Identifications obtained by the above-mentioned methods were compared to those attained from colonies grown after culturing the broth from the same BCs on Columbia blood-agar plates (bioMérieux, France). The colonies were identified using on-plate protein extraction with 70% formic acid [1].”

  1. How were cases with scores below 1.7 with all the top hits being the same handled? Reported to the genus level?

In those cases, the top three identification provided by MALDI-TOF were compared to the identification obtained from colonies (gold-standard method). If both identifications are concordant at the species-level, the low-score identification was considered at this level. Otherwise, it was only considered at genus-level. In none of the cases there were disagreements at genus-level. This fact has been explained in Lines 159-164: “Isolates identified with score values below 1.6 were considered only when the first three identifications provided by MALDI-TOF MS were consistent either at the species- or the genus level. Otherwise, the identification was considered “not reliable”. Agreement at species- and genus level between the methods tested and the gold standard (identification from isolated colonies) was also analyzed.”

Reviewer 2 Report

In the manuscript entitled “Evaluation of the rapidBACpro® II kit for the accurate identification of microorganisms directly from blood cultures using MALDI-TOF MS” the Authors have described the identification of microorganisms approach directly from blood cultures using MALDI-TOF MS.

shown to be the most impacting application of this methodology.

This paper should be revised according to the following comments.

  1. In the introduction, part Authors should add extended information about the utilization of blood culture kits for dedicated bacteria strains, e.g. anaerobic, facultative anaerobe.
  2. The main issue of work. Why authors, before the performed study did not test the proposed kit with reference clinical relevant bacteria, yeast strain?
  3. What about the utilization of another matrix in the case of yeast identification? Did Author try utilization of DHB? Please, perform additional discussion.
  4. Please, discuss obtained statistical data in the context of specificity of bacterial strain for used kits.

In my opinion, the manuscript required a revision before final acceptance. 

Author Response

Reviewer 2

This paper should be revised according to the following comments.

1. In the introduction, part Authors should add extended information about the utilization of
blood culture kits for dedicated bacteria strains, e.g. anaerobic, facultative anaerobe.

The requested information has been added in Lines 65-66: Obligate and facultative anaerobic species have also shown to be identified using this kit [12, 14]. Unfortunately, there is no available information about the specific performance of other kits for the identification of anaerobic species.

2. The main issue of work. Why authors, before the performed study did not test the proposed kit with reference clinical relevant bacteria, yeast strain?

When the multicentric study was organized, the authors had previous information about the performance of the kit because of the studies from Ashizawa et al. (PMID: 28461023) and Tsuchida (PMID: 30075236) were already published and the paper from Kayin (PMID: 31494828) was available as a pre-accepted manuscript. Thanks to these studies we knew that the kit provided reliable results for the most frequently encounter species. The purpose of our multicentric study was to evaluate the performance of the kit but also its reproducibility in different laboratories.

3. What about the utilization of another matrix in the case of yeast identification? Did Author try utilization of DHB? Please, perform additional discussion.

In our setting, DHB is used for the analysis of high molecular weight molecules. The use of -Cyano-4-hydroxycinnamic acid (HCCA) for the identification of microorganisms is widely applied and provides highly accurate results for proteins between 2 and 20KDa.

4. Please, discuss obtained statistical data in the context of specificity of bacterial strain for used kits.

In the reviewed manuscript (Lines 332-335) we state that: In our study, 761 monomicrobial BCs were analyzed in four laboratories using the rapidBAC pro® II kit. We identified 80.0% of the isolates correctly at the species level with score values ≥2.0 and 94.4% correctly at the genus level with score values ≥1.7. The re-maining isolates (n=42) could be reliably identified despite the lower scores. Later on, in lines 363-365 we say that However, no misidentifications with any of the applied methods have been detected in this study, showing that the methodology applied is highly specific.

In my opinion, the manuscript required a revision before final acceptance.

The style and the content of the manuscript has been thoroughly reviewed
